# Velocities of hippocampal traveling waves are proportional to their coherence frequency

**Gadi Goelman** [1,2*°], **Tal Benoliel**[1,2°], **Zvi Israel**[2,3], **Sami Heymann**[3], **Juan Leon**[3], **Dana Ekstein**[1,2]

**1** Department of Neurology and Agnes Ginges Center for Human Neurogenetics, Hadassah Medical Center, Jerusalem, Israel, **2** Faculty of Medicine, The Hebrew University of Jerusalem, Jerusalem, Israel, **3** Department of Neurosurgery, Hadassah Medical Center, Jerusalem, Israel

° These authors contributed equally to this work.

* gadig@hadassah.org.il

## Abstract

Cortical traveling waves, defined by their spatial, temporal, and frequency characteristics, provide key insights into active brain regions, timing, frequency, and the direction of activity propagation. Emerging evidence suggests that the directionality and spatiotemporal extent of these waves encode cognitive processes. However, the relationship between frequency and this encoding mechanism remains unclear. We investigate the hypothesis that coherence frequency determines wave propagation velocity. By employing both bivariate linear and multivariate nonlinear coherence analyses, we demonstrate that coherence frequency encodes propagation velocity. Unlike linear analyses, which may overestimate velocities due to bidirectional flow when assessing multiple pair coherences, our nonlinear approach—calculating propagation along four-node pathways—treats pathways as holistic units with net unidirectional flow, making it more appropriate for calculating wave velocities. We extracted pairwise coherence and four-node pathways from local field potentials recorded via intracranial electrodes positioned along the hippocampal longitudinal axis in patients with drug-resistant epilepsy. Our findings reveal that average coherence values and contact pair distances calculated by the multivariate analysis are more consistent across frequencies compared to pairwise coherence. The average coherence values are higher, and the average pair distances and wave velocities are lower in the multivariate analysis than in the pairwise approach. Propagation velocities along the hippocampus at low frequencies (<~35 Hz) exhibit a linear dependence on frequency in the alpha and beta bands, with a steeper slope in the gamma band, indicating distinct mechanisms for velocity-frequency dependence across oscillation bands. While observed within the hippocampus, these findings suggest that the relationship between frequency and wave velocity may extend to other cortical areas. Our nonlinear multivariate analysis appears better suited than pairwise coherence for investigating brain network dynamics. Further research is needed to elucidate the role of conduction velocity in brain function.

**Data availability statement:** Data sets are available upon request and after approval by the Hadassah ethics committee. The reason is that it potentially contains sensitive patient information. For information contact the Hadassah Helsinki committee. email: vhelsinli@hadassah.org.il phone: 972-02-6777242

**Funding:** The author(s) received no specific funding for this work.

**Competing interests:** The authors have declared that no competing interests exist.

## 1. Introduction

Traveling waves that are based on coherence [1,2], have been observed in multiple regions in humans[3,4] and animals [5,6], at both small and large scales. They have been suggested to elucidate the coordination of neural activity between regions, the mechanism of large-scale neural organization [1,2], a mechanism for the "correspondence problem" [7], and the brain's ability to dynamically adapt to behavioral and cognitive changes[8,9]. Recently, it has been demonstrated that in episodic memory, measured directly in humans, the directions of propagations encode different memory processes, suggesting a link between directionality and the timing of regional activity with cognition and behavior [10]. Since traveling waves frequency, spatial and temporal aspects are thought to indicate which region is active, at what time, at what frequency, and in which direction activity is propagating, it is crucial to understand what determines the velocity of propagation.

Inspection of time-frequency wavelet plots of coherence between local-field-potential time-series signals consistently revealed that the width of coherence windows, assumed to represent periods during which communication takes place[11,12], exhibits broader characteristics for lower frequencies and narrower features for higher frequencies. This consistent observation strongly implies a proportional relationship between the size of time windows and the associated wavelength.

This suggests that communication facilitated by coherence is governed by the duration of these time-frequency windows, thus exhibiting frequency-dependent characteristics. Consequently, following on findings reported by others [13–15], we hypothesize that coherence frequency dictates the velocity of traveling wave propagation, so that higher frequencies correspond to faster propagations, whereas lower frequencies correspond to slower propagations.

To test this, we conducted an analysis of traveling waves within local field potential (LFP) signals extracted from the human hippocampus in neurosurgical epileptic subjects undergoing stereo-electroencephalography (SEEG) electrode implantation. In contrast to previous studies employing various bivariate linear methods for measuring traveling waves, we utilized a novel multivariate nonlinear analysis method previously applied to functional MRI signals [16–22]. This nonlinear analysis leverages mutual coherence (simultaneous coherence) among four time-series (i.e., LFP signals, each from a distinct location), along with the temporal delays at each frequency between these signals, to establish their temporal order and infer directed pathways. This approach is particularly suitable for studying traveling waves as it treats pathways as holistic units rather than mere combinations of their constituent parts and assumes effectively unidirectional flow along these pathways.

Utilizing coherence to estimate transmission times and velocities relies on the assumption that the time of transmission is proportional to coherence phase over frequency, representing the time-lag between regions. However, the bidirectional nature of many cortical and hippocampal connections can lead to coherence with zero or close to zero phase lag (zero-lag synchrony) [23]. Consequently, employing linear analyses by considering multiple pair coherences, each including a mixture of connections, to track wave propagation, may lead to underestimated transmission times and, consequently, overestimated velocities. In contrast, the calculation of pathways by multivariate analysis assumed effectively unidirectional flow between its regions thus more suitable for accurately measuring transmission times from coherence phases. Results from both linear and nonlinear analyses are compared to elucidate and illustrate this issue.

## 2. Methods

### 2.1. Subjects

The study involved three subjects with drug-resistant epilepsy undergoing intracranial EEG monitoring, including depth electrodes inserted along the hippocampal longitudinal axis as

part of their pre-surgical evaluation. The study was approved by the Ethics Committees of the Hadassah Medical Center, Jerusalem, Israel. Electronic medical records, imaging and EEG studies of the patients were accessed by the authors (TB, SH, JL and DE) on March 8-9th and May 29th 2023, and on January 10th, April 16th and May 26th 2024. The relevant parts of data were chosen and identified prior to further processing. Table 1 provides a comprehensive overview of the subjects' clinical, imaging, and electrophysiological data.

## 2.2. SEEG data acquisition and calculations

Patients underwent implantation with depth electrodes (DIXI 15 contacts, interval 3.5 mm, diameter 0.8 mm), with the most anterior contact positioned first (numbered 1). In addition, scalp EEG, using 21 electrodes placed in accordance with the 20-10 system, was acquired. Data were collected using the XLtec clinical system (Natus Medical Inc.) sampled at 2048 Hz.

To ensure the accurate localization of electrode contacts within the hippocampus, a specialized neurosurgeon used merged CT/MRI images to identify and document all contact positions, distinguishing whether they were within or outside the hippocampal region. S1–S5 Figs display exemplar images derived from these merged CT/MRI images. Additionally, we selected data recorded at least 1 minute before and after seizures, free from abnormal inter-ictal activity, as determined through visual inspection of the ECoG and video recordings by

**Table 1. A comprehensive overview of the subjects' clinical, imaging, and electrophysiological data.**

|  | Pt. 1 Rt. hip | Pt. 1 Lt. hip | Pt. 2 Rt. hip | Pt. 2 Lt. hip | Pt. 3 Rt. hip |
|---|---|---|---|---|---|
| Atrophy | + (MTS) | – | – | – | + |
| Seizures | + | – | + | + | + |
| Inter-ictal activity | + | + | + | + | + |
| Gender | Male |  | Female |  | Female |
| Age of onset/epilepsy duration (years) | 15/20 |  | 25/15 |  | 17/4 |
| Seizure semiology | Lt hand tingling, dialeptic, lt clonic |  | Anxiety, dialeptic |  | Lt. hand tonic, dialeptic |
| EEG | ii: Right and left temporal sharp waves Ictal: diffuse onset, sometimes more pronounced right temporal |  | ii:Left temporal slowing and sharp waves ictal: independent left and right frontotemporal onset |  | ii: right parietotemporal spikes ictal: right posterior temporal |
| MRI | Right frontal cystic changes, right MTS |  | normal |  | Right frontal porencephaly, rt MTS |
| PET CT | Rt. frontotemporal hypometabolism |  | Rt. frontal and lt. mesial temporal hypometabolism |  | Right frontotemporal hypometabolism |
| Neuropsychological | N/A |  | Memory and executive functions impairments |  | Visual memory impairment, mild concentration difficulty and mild decrease in cognitive processing |
| FMRI language lateralization | Bilateral |  | Bilateral, left more than right |  | Left |
| Clinical hypothesis prior to SEEG | Independent bitemporal versus rt. temporal lobe epilepsy |  | Independent bitemporal seizures versus lt. temporal neocortical |  | Rt. hippocampal seizures, possible dual pathology |
| Clinical SEEG seizure onset | Rt. hippocampus. |  | Lt. temporal neocortex, rt. hippocampus. |  | Rt. hippocampus (24/27); unclear, probably rt. frontal (3/27 seizures) |
| ASMs | CBZ, CLB |  | CBZ, BRV, CLB, cannabis |  | CBZ, LVT, SOS clonazepam |
| Past ASMs | PHT, LEV |  | LTG, LEV, CZP, permapanel; VNS |  | LCM, VPA |
| Surgical intervention | Rt. ATL + AH |  | RNS (rt. and lt. hippocampus) |  | Rt ATL+AH |
| Pathology | Hippocampal sclerosis |  | N/A |  | Hippocampal sclerosis, focal cortical dysplasia IIIa |
| Post-procedure outcome | Seizure free |  | Seizure free |  | Seizure recurrence |

Rt. Right, Lt. Left, MTS mesial temporal sclerosis, ASM ani seizure medication, PHT phenobarbital,LEV levetiracetam, CLB clobazam, CZP clonazepam, CBZ carbamazepine, LCM lacosamide, RNS responsive neurostimulation, ATL anterior temporal lobectomy, AH amygdalohypocampectomy

an expert epileptologist. The calculations of pathway indexes and phase lag indexes were performed using 500 time-windows (referred to hereafter as a 'set'), each lasting 1 second. To account for mood and other temporary health variations, eight or more sets were independently analyzed to calculate the pathway indexes and phase lag indexes, which were then averaged.

Specifically, we employed 500 time-windows, each spanning 1 second and containing 1024 time points. The time series (local field potentials recorded from each electrode contact) were first transformed into the time-frequency domain using wavelet transformation. The wavelet coefficients were then averaged over time, and the phase was calculated.

To assess the impact of spikes, we conducted simulations with spiked data and repeated the aforementioned calculations. Our results demonstrated that the spikes fell outside the wavelet confidence region and were not included in the calculations, confirming that the findings were not influenced by wavelet-filtering artifacts [24,25].

To avoid bias introduced by the selection of specific time series, all possible 105 pairs were calculated for the pair coherence, and since each pathway calculation involved four contact signals, all possible combinations of four signals out of the 15 available contact signals resulting in a total of 1365 combinations were calculated. This approach ensured an equal chance of occurrence for all contacts, pair coherences and pathways.

## 2.3. Software

For the bivariate (pair coherence) and the multivariate (four-node pathways) calculations, we used wavelet analysis with custom-developed software in IDL version 8.2.0 (Exelis Visual Information Solutions, Inc.). Wavelet analysis was selected since it decomposes the time-signals into time-frequency components, making it most suitable for analyzing velocities as function of frequency. The wavelet software was provided by Torrence and Compo [26,27]. Calculations were performed on 1sec long time-windows, each with 1024 points after down sampling the data to 1KHz. The complex Morlet wavelet functions, which have been demonstrated to offer a favorable balance between time and frequency localization [28], were used. In the calculations, we set the smallest scale to 0.01, 1 millisecond for the time resolution and employed 31 scales to cover a frequency range of 7.2-96.8 Hz. This frequency range was selected to include the alpha to high gamma bands but excluded the theta frequencies due to the time windows' length.

## 2.4. PLI calculations

The Phase Lag Index (PLI) computation followed the methodology outlined by Stam et al [29,30]. Initially, time-series (contact signals), each spanning 1 second with 1024 points, underwent wavelet transformation to transition into the time-frequency domain. Subsequently, pair phase coherence was calculated as a function of time and frequency for all possible 105 pairs. As spontaneous activity data is not synchronized to any specific event, the product of the wavelet coefficient of one time-series with the complex conjugate of the wavelet coefficients of the other time-series (that is the definition of coherence) was averaged over time. The pair coherence for the *i-j* pair was then calculated using the formula:

$$\Delta\varnothing_{i,j}(\nu) = \mathrm{atan}\overline{(W_i(\nu,t) \cdot W_j^*(\nu,t))}. \tag{1}$$

This process was iterated over 500 time-windows for each pair, and the phase coherence was determined for each window. PLI was subsequently calculated to define significant coherence for each pair over these 500 windows as:

$$PLI_{i,j}(\nu) = \left| \frac{1}{500} \sum_{k=1}^{500} sign\left(\Delta\varnothing_{i,j}^{k}(\nu)\right) \right| \tag{2}$$

In addition, we computed the average phase-coherence and the average contact distance (in cm) over significant coherences (average over $l$ significant pairs) for each frequency:

$$\Delta\varnothing(\nu) = \text{atan}\left(\sum_{l=1}^{n}\sum_{k=1}^{500} W_i(\nu,t)\cdot W_j^{*}(\nu,t)\right) \tag{3}$$

and

$$Dis(\nu) = \frac{1}{n}\sum_{l=1}^{n}|i-j|*0.35) \tag{4}$$

where $n$ is the number of significant coherences at frequency $\nu$. Finally, velocity is calculated over significant pairs as:

$$V(\nu) = \frac{Dis(\nu)\cdot\nu}{\Delta\varnothing(\nu)} \tag{5}$$

## 2.5. Pathway index (PWI) calculations

In our prior study, we showcased the utilization of interactions among four coupled time-series to determine their temporal coupling order at each time-frequency point [16–22]. In these calculations we assumed that propagation was predominantly unidirectional. The fundamental analysis framework for the four coupled time-series has been previously outlined and will be briefly summarized here for clarity including a hypothetical example.

The conditions for coherence between four time-series can be described by the mutual existence of three expressions, which we referred to as functional connectivity (FC) [6]:

$$FC^1(\nu,t) = W_{\nu,t}^{j1}\cdot(W_{\nu,t}^{j2})^{*}\cdot W_{\nu,t}^{j3}\cdot(W_{\nu,t}^{j4})^{*} = A\cdot\left\{\exp\left[i\left(\vartheta_{\nu,t}^{j1} - \vartheta_{\nu,t}^{j2} + \vartheta_{\nu,t}^{j3} - \vartheta_{\nu,t}^{j4}\right)\right]\right\} = Ae^{i\varphi^a}$$

$$FC^2(\nu,t) = W_{\nu,t}^{j1}\cdot W_{\nu,t}^{j2}\cdot(W_{\nu,t}^{j3})^{*}\cdot(W_{\nu,t}^{j4})^{*} = A\cdot\left\{\exp\left[i\left(\vartheta_{\nu,t}^{j1} + \vartheta_{\nu,t}^{j2} - \vartheta_{\nu,t}^{j3} - \vartheta_{\nu,t}^{j4}\right)\right]\right\} = Ae^{i\varphi^b} \tag{6}$$

$$FC^3(\nu,t) = W_{\nu,t}^{j1}\cdot(W_{\nu,t}^{j2})^{*}\cdot(W_{\nu,t}^{j3})^{*}\cdot W_{\nu,t}^{j4} = A\cdot\left\{\exp\left[i\left(\vartheta_{\nu,t}^{j1} - \vartheta_{\nu,t}^{j2} - \vartheta_{\nu,t}^{j3} + \vartheta_{\nu,t}^{j4}\right)\right]\right\} = Ae^{i\varphi^a}$$

with, $W_{\nu,t}^{j1}$ and $\vartheta_{\nu,t}^{j1}$ the wavelet coefficient and phase of time series '$j1$' at a time-frequency point, where the numbers 1 to 4 represent the four time-series. A' the amplitude, * denotes complex conjugate and '$i$' the imaginary unit. Pair like phase-differences were expressed by the $\varphi^a$, $\varphi^b$, and $\varphi^c$ phases as:

$$\Delta\theta_{2-1}(\nu,t) = \frac{1}{2}(\varphi^a + \varphi^c); \Delta\theta_{3-1}(\nu,t) = \frac{1}{2}(\varphi^b + \varphi^c); \Delta\theta_{4-1}(\nu,t) = \frac{1}{2}(\varphi^a + \varphi^b)$$

$$\Delta\theta_{2-3}(\nu,t) = \frac{1}{2}(\varphi^a - \varphi^b); \Delta\theta_{4-3}(\nu,t) = \frac{1}{2}(\varphi^a - \varphi^c); \Delta\theta_{4-2}(\nu,t) = \frac{1}{2}(\varphi^b - \varphi^c). \tag{7}$$

Similar to pair coherence for spontaneous activity data, we average the phases over time, within each 1sec time window. These averaged phases are then utilized to determine the temporal order, referred to as pathways, at each frequency.

Our focus lies on effectively continuous and unidirectional pathways, indicating sequences that originate from one ensemble and sequentially pass through all other ensembles. For four-node pathways, there exist a total of 24 different pathways, as listed in S1 Table.

To establish pathways independent of the reference phase and the $2\pi$ limitation (as described in [9,10], see example below), we define pathways only if all four phase differences, each with a different reference phase, yielded the same pathway that is, the same temporal order. Specifically, we defined a pathway's index using the following criteria:

$$PWI_l^k(\nu) = \frac{1}{500}\sum_{i=1}^{500} \begin{cases} 1 \\ 0 \end{cases} \begin{array}{l} \textit{phases in line with the l pathway for all 4 reference phases} \\ \textit{phases not in line with the pathway} \end{array} \tag{8}$$

with '$l$' = 1, 2... 24 corresponding to a pathway's number in S1 Table, $k$ refers to the choice of the 4 time-series (out of the 1365 possibilities), and summation is over the 500 time-windows. We note that averaging of wavelet-coefficients over time-windows solves the intrinsic time-frequency uncertainty [26,27].

Next, we calculated the average pair distances for pairs within pathways for significant pathways similar to Equation 4. We then determined the average pair distance along each significant pathway, calculated as one-third of the sum of the three pair distances. Similarly, the average phase of a pathway was calculated as one-third of the sum of the absolute values of its three phases, and velocity was obtained using Equation 5 with the corresponding distance and phase of the pathway.

## 2.6. An example

To clarify how pathways are calculated, consider the following example (see Fig 1, which illustrates only the amplitude values). Given four LFP signals from different contacts of an SEEG electrode implanted along the hippocampus of a subject, one of the 1365 possible combinations of the 15 signals from the 15 contacts is selected (Fig 1, top panel). First, each signal is expanded in wavelet space for a given 1-second time window, covering frequencies up to 100 Hz (Fig 1, middle panel, showing only the lower frequencies).

Next, we apply Equation 6 to compute the mutual coherence among all four signals (Fig 1, bottom-left). As shown, mutual coherence defines time-frequency windows during which simultaneous interactions between all four regions occur, allowing for information exchange. Since the signals during resting-state are not synchronized to any specific event, we average the wavelet-transformed signals over time, yielding a function of frequency (Fig 1, bottom-middle).

At each frequency, Equation 7 is used to define pairwise coherences, which are then compared to identify the pathways that match the order of these coherences (listed in S1 Table). For instance, at a specific frequency, $\nu = 10Hz$, the pairwise coherences might be as follows: $\Delta\theta_{2-1}(\nu) = 50^0$, $\Delta\theta_{3-1}(\nu) = 80^0$, $\Delta\theta_{4-1}(\nu) = 20^0$. These coherences suggest the temporal order '1-4-2-3,' one of the 24 possible orders listed in S1 Table.

To avoid errors due to phase differences exceeding $2\pi$, the calculation is repeated three more times using different reference signals. A valid temporal order is only accepted if it remains consistent across all reference choices. Note that in some cases, a pathway for specific signals cannot be defined. This process is repeated 500 times, each on a different time window, to ensure statistical significance as defined by Equation 8.

To determine directed pathways, we assume that a positive phase difference between regions i and j indicates information transfer from i to j or vice versa. However, for the current study, determining the temporal order of interactions is sufficient.

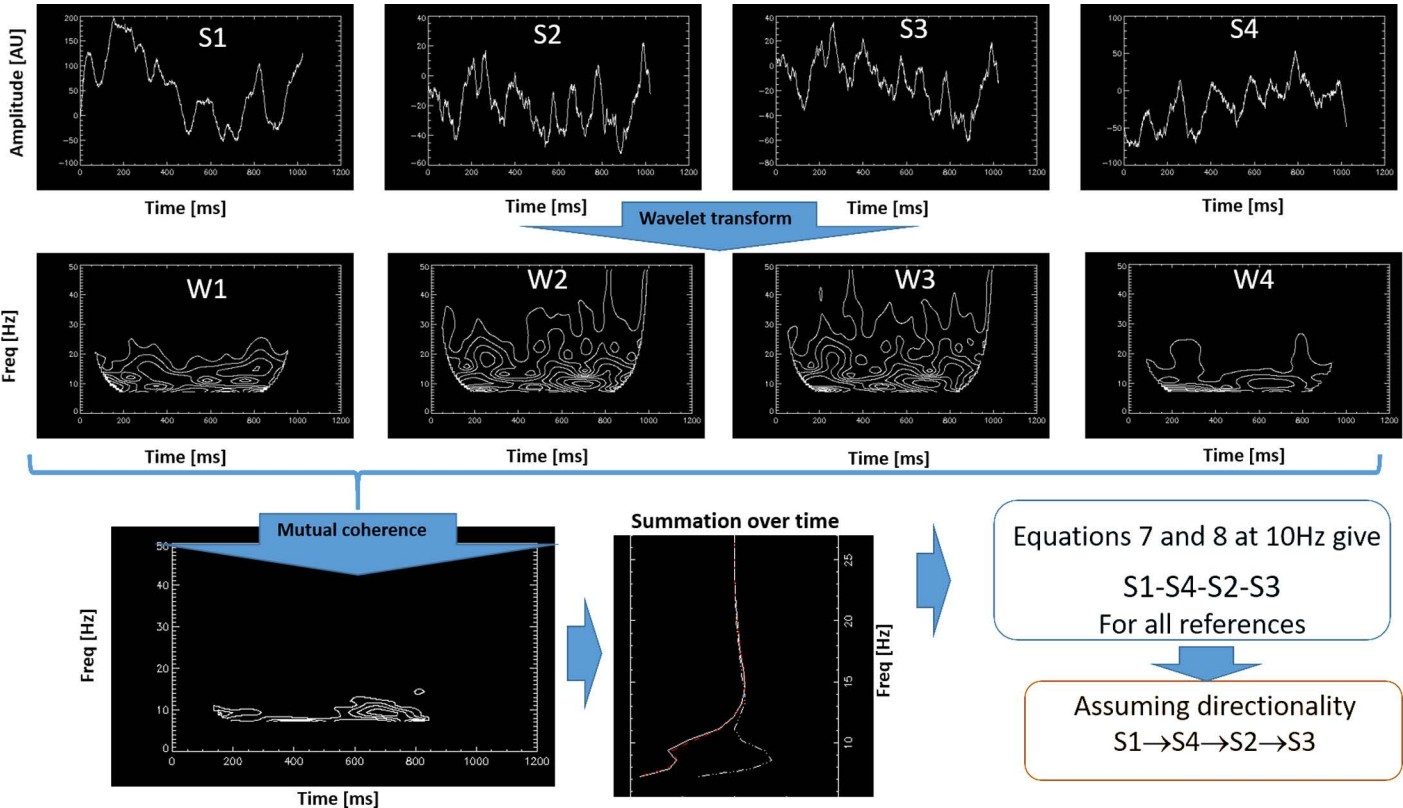

**Fig 1. Illustration of the steps to obtain a pathway.** A pathway is calculated for a set of four LFP signals, each from a different location along the hippocampus (top panel). Each signal is wavelet-transformed into time-frequency space (middle panel, showing only amplitudes). These wavelet coefficients are combined using Equation 6 to define time-frequency windows of mutual coherence, which represent periods where information can be exchanged among the four regions (bottom-left). During resting-state, we average over time to obtain a frequency-dependent function (bottom-middle). Equations 7 and 8 are then applied to define the temporal order of the signals, and a directed pathway is determined using directionality criteria.

## 2.5. Statistical analysis

We employed permutation non-parametric tests to compute the null distributions of PLI (Eq. 2) and PWI (Eq. 8) separately for each frequency. To ensure the uncoupling of the 15 contact signals, a random number generator was utilized to select time-windows from which the contact signals were obtained. For the PLI calculations, data from subject 2 right hippocampus were used, and null calculations were repeated 10,000 times with PLI calculated on each iteration. For the PWI calculations, data from subject 1 left hippocampus were used, and this process was repeated 100,000 times with Equation 8 calculated on each iteration. Due to low null values in the PWI calculations, a higher number of iterations was needed.

In the null PLI calculations, a value of 0.19 corresponded to $< 1.6 \cdot 10^{-5}$. To correct for multiple comparisons involving 105 * 31 combinations and frequencies, even using the most conservative Bonferroni correction, a PLI value of 0.19 corresponded to a corrected p-value of 0.05, and this value was used in the calculations.

For the PWI calculations, the highest value obtained for PWI was 0.057. The null distributions across all frequencies indicated that a PWI value of 0.05 corresponded to $p < 10-5$. To account for multiple comparisons involving the 1,365 * 31 combinations of pathways and frequencies, we implemented a higher cutoff value. Since the null distributions could not estimate p-values for PWI values higher than 0.056 (even with 100,000 calculations), we set a

conservative cutoff of 0.15. This cutoff value ensured that randomized null signals yielded zero pathways, while correct data resulted in only a few hundred pathways at specific frequencies.

## 3. Results

Fig 2 displays the average coherences (mean values ± standard errors computed across sets) across frequencies for all five hippocampi, focusing on significant pathways (Fig 2A) and significant pairs (Fig 2B). Note that the number of significant pathways and pairs vary between subjects. These values and their dependence on frequency is given in S6 Fig. As shown in Fig 2, average coherence values of pathways calculated by the multivariate analysis were consistently higher compared to pair coherences calculated by bivariate analysis, across all frequencies. Additionally, average coherence values of pathways across different hippocampi were similar, remaining approximately constant at low to intermediate frequencies, before declining at frequencies above ~ 35 Hz. In contrast, pair coherences were lower, exhibited greater variability between hippocampi, and did not show frequency dependence.

Fig 3 presents the average pair distance between electrodes pairs (mean ± SE) for pairs within pathways (Fig 3A) and pairs calculated by the PLI (Fig 3B) across frequencies. Average pair distances within pathways were lower than pair distances calculated by bivariate analysis and were relatively constant across frequencies, with low variability except for the hippocampus of subject 3, whose distance at high frequencies was larger. Conversely, distances for bivariate pair coherences were higher and showed significant variation between hippocampi. In this case, the pair distance for the hippocampus of subject 3 was an outlier, with higher values that increased with frequency.

Fig 4 illustrates the calculated average velocities within pathways and for pair bivariate coherences, with values higher by one order of magnitude for the later (note the difference in scale). The velocities for all hippocampi, in both bivariate and multivariate calculations, showed a moderate increase with frequency for alpha to beta bands, followed by a sharp rise at gamma frequencies. For the pathway calculations, this trend showed a linear dependency of velocity on frequency in the alpha and beta ranges. The velocity followed an approximately

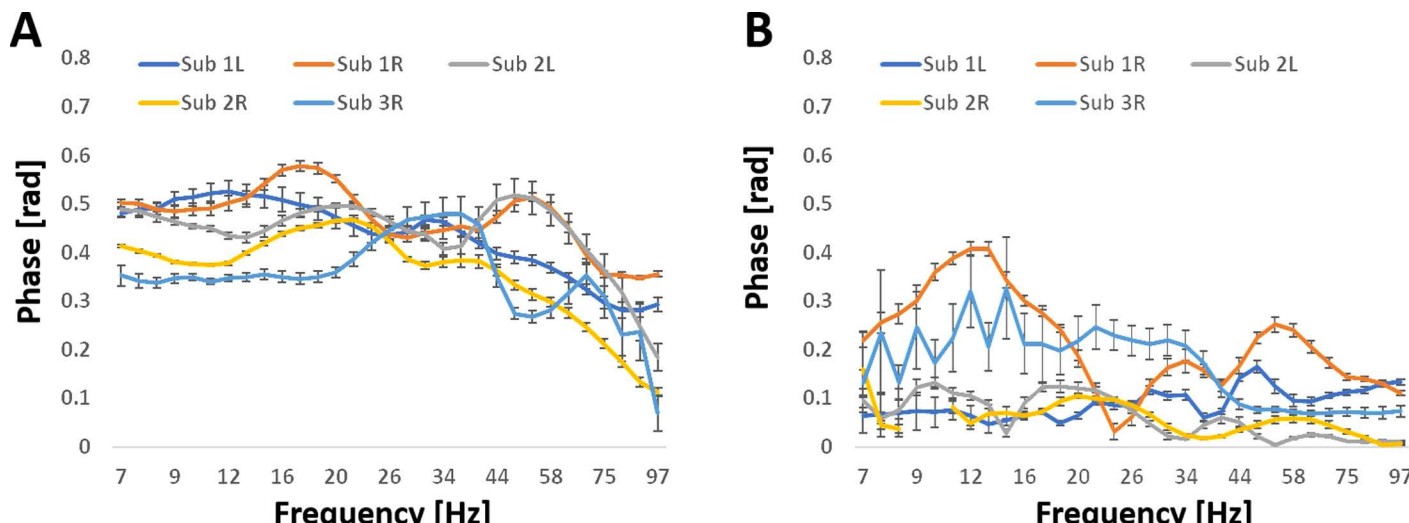

**Fig 2. Coherences of the five hippocampi used in the analysis as a function of frequency.** For each hippocampus, the mean values ± standard errors computed across different sets are presented. A. Average coherences over the three pairwise connections within significant four-node pathways, calculated using multivariate nonlinear analysis. B. Average coherences over all significant pairs, calculated using bivariate linear analysis.

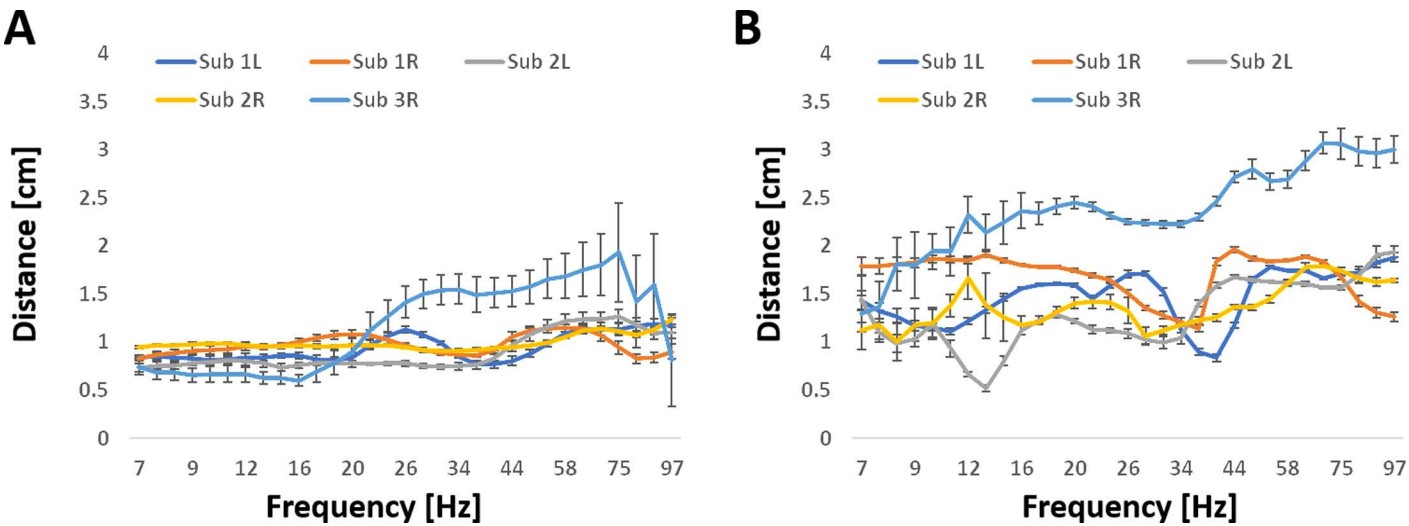

**Fig 3. Coupling distances of the five hippocampi used in the analysis as a function of frequency.** For each hippocampus, the mean values ± standard errors computed across different sets are presented. A. Average distances (difference between contacts) over the three pairwise connections within significant four-node pathways, calculated using multivariate nonlinear analysis. B. Average distances over all significant pairs, calculated using bivariate linear analysis.

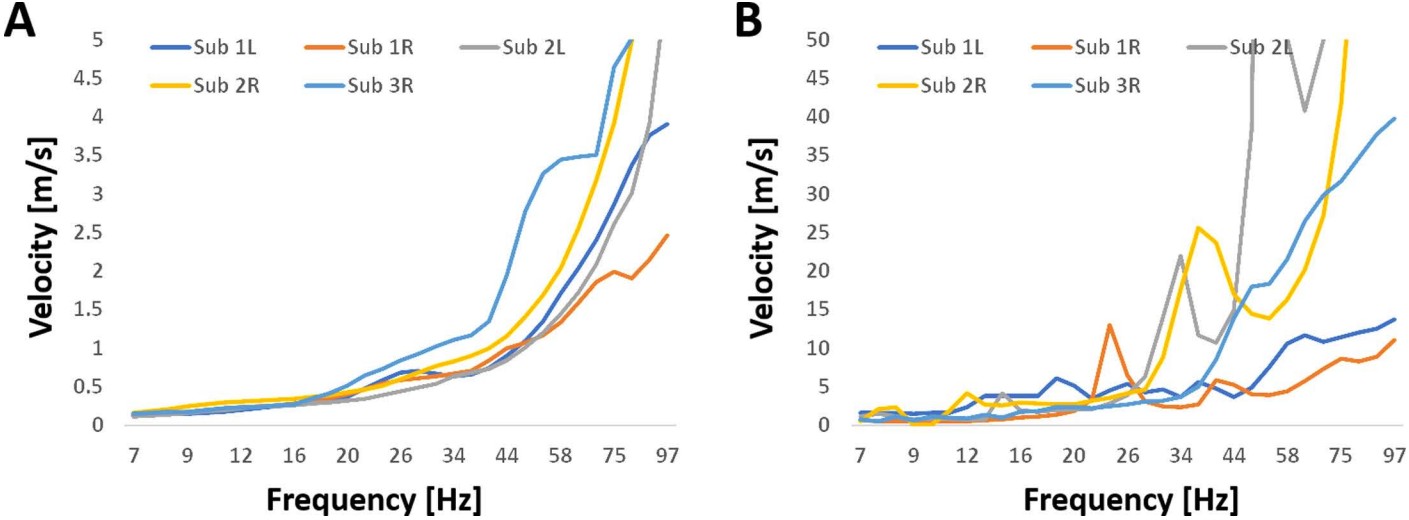

**Fig 4. Velocities of the five hippocampi used in the analysis as a function of frequency.** Velocity was calculated using the average distance and coherence. A. Average velocity over the three pairwise connections within significant four-node pathways, calculated using multivariate nonlinear analysis. B. Average velocity over all significant pairs, calculated using bivariate linear analysis. Note the 10-fold higher y-axes.

linear dependency: $V(\nu) \sim 0.02 \cdot \nu$ where the constant is in units of meters. This gives a spatial frequency of $50\frac{1}{m}$ and a spatial wavelength of 2 cm, approximating the pathway's length. The frequency at which the linear dependency transitioned to a sharp increase varied between hippocampi: approximately 35 Hz for both hippocampi of subject 1, 20 Hz for the right and 44 Hz for the left hippocampus of subject 2, and around 20 Hz for the hippocampi of subject 3. These values suggest a moderate linear dependency of velocity on frequency for the alpha to beta bands, with a sharper dependency for the gamma band.

## 4. Discussion

This study explores the hypothesis that travelling wave velocity is proportional to the wavelength of oscillation. Our key finding, that information transfer velocity correlates with frequency, shines the spotlight on travelling wave velocity rather than frequency, and opens up the question of how different transfer velocities may affect communication between pathway nodes, and even network architecture.

While the dependence of velocity on frequency has been previously demonstrated, we argue that using four-node pathways calculated by multivariate analysis instead of bivariate pairwise coherence offers a methodological advantage in measuring information transfer velocity. Given the expected bidirectional communication present in most hippocampal and cortical connections, employing bivariate pair calculations to measure the average coherence between all pairs at each frequency could result in an overestimation of velocity. This overestimation occurs because the average pairwise coherence at a specific frequency includes local and remote pairs with varying coherence values, including coherence of zero or close to zero [23]. In contrast, four-node pathways are specifically designed to support effective unidirectional transmission along pathways. Although communication along these pathways can be bidirectional, the forward direction, by definition, is stronger; otherwise, the pathways would not be defined (i.e., the pathway index in Equation 8 would be zero). This means that pathways are not defined if the pair-coherence along the pathways equals zero, effectively excluding cases of zero coherence. Nevertheless, velocity estimation even with the multivariate analysis could still be overestimated, but this overestimation is likely to be less than that derived from bivariate pairwise coherence calculations.

Analyses were performed on SEEG electrodes signals collected from the hippocampi of neurosurgical epileptic subjects. Specifically, the data were derived from depth electrodes strategically placed along the long axes of the hippocampus, providing an optimal dataset for testing the proposed hypotheses. The choice of the hippocampus was motivated by its anticipated one-dimensional flow, enabling the estimation of transmission distances and consequently, the calculation of transmission velocities. Electrode placement allowed for the collection of data from five hippocampi in three subjects, each recorded for approximately three days. This extended recording period enabled averaging over multiple time-windows, minimizing the impact of uncontrolled fluctuations on the analysis. The choice of spontaneous activity (resting-state) data over stimulated data, with the latter tuned to frequencies operated by the specific stimulation used, guarantees a broad band of frequencies that are expected during spontaneous activity. The fact that similar findings were observed in all five hippocampi, regardless their different health state, implies that the linkage between frequency and velocities reflects a physiologic, rather than pathologic, attribute of hippocampal information processing.

Our primary findings reveal that the velocity of transmission, calculated by pair and by four-node coherences, increases with frequency for all hippocampi. However, and as anticipated, the values are much higher for bivariate pair coherence and likely are biased by bidirectional communications and zero phase coherences. In contrast, velocity values obtained by the multivariate analysis are in line with previous publications and demonstrate different dependencies for low and high frequencies. This proposition aligns with the prevailing understanding that there is no singular mechanism dictating communication speed in the brain. Instead, communication is influenced by a combination of factors, including the type of connection, myelination, synaptic transmission, and neural oscillations frequencies. Therefore, while our proposal suggests a general principle of communication through neural oscillations, various mechanisms may be in play influencing the different results obtained for lower (<35Hz) and higher (>35Hz) frequencies.

Regarding the speed of wave propagation, using EEG simulation, stable wave propagation velocity was observed within a biologically plausible range akin to axonal conduction speeds

of 1–10 m/s [26]. In the prefrontal cortex of monkeys, the velocity of traveling waves within theta, alpha, and beta frequency ranges exhibited a linear increase with frequency, ranging from approximately 0.2 m/s to 0.5 m/s [14]. Using 2D phase-based methods, the velocity of traveling waves in the visual cortex of monkeys at frequencies 5-20Hz was estimated to be 0.25-1.35m/s [15]. Similarly, a study on memory processing in humans at approximately 9 Hz, cortical propagating waves were found to travel at approximately 1 m/s[10]. These trends are consistent with our findings and the values obtained through multivariate analysis, as illustrated in Fig 4.

Our observation of a sharp increase in velocity for high frequencies (>35 Hz) suggests that low and high frequencies exhibit distinct dependencies on frequency. This observation aligns with the prevailing notion that neuronal oscillations in low versus high frequencies involve different underlying mechanisms. While oscillations at low (<30 Hz) frequencies are deemed crucial in neuronal communication and processing [31,32], high-frequency oscillations are primarily attributed to local activity, often stemming from broadband multi-unit spiking activity [33,34]. Consistent with these disparities, studies have shown that memory encoding and retrieval processes in whole-brain human connectivity tend to desynchronize for fast gamma (30–100 Hz) and synchronize for slow theta (3–8 Hz) during encoding and retrieval [35]. Moreover, it has been demonstrated that trial-by-trial fMRI BOLD fluctuations positively correlate with trial-by-trial fluctuations in high-EEG gamma power (60–80 Hz) and negatively correlate with alpha and beta power, suggesting distinct mechanisms for different oscillation frequencies [36].

Our results may provide insight into the temporal dynamics governing brain circuits, and raise the question of how varying velocities of information transfer within a network may influence the network's computational qualities. For instance, using magnetoencephalography human data at resting state, a loop composing of posterior-to-anterior information flow dominated by regions in the visual cortex and posterior default mode network was observed at higher frequency bands (alpha and beta), while an anterior-to-posterior flow, involving mainly regions in the frontal lobe that were sending information to a more distributed network, was described in the theta band [37]. Similarly, studies in monkeys have demonstrated that theta and gamma oscillations in the visual system promote information flow in the feedforward direction during bottom-up processing, while beta oscillations facilitate feedback interaction during top-down processing [38]. Building upon our findings, we interpret these observations as comprehensive feedback cycles, encompassing a faster bottom-up flow facilitating processing in higher regions, alongside a slower top-down flow, enabling balanced bottom-up and top-down circuits.

## Limitations

Our study utilized data collected from unhealthy hippocampi, each affected by varying degrees of disease duration and severity. While efforts were made to select data exhibiting no abnormal activity, it remains possible that the ostensibly 'normal' appearing LFP may possess some abnormal characteristics. We did not employ methods such as regression to mitigate differences between subjects and hippocampi, as we were concerned about potential biases of such regressions and due to the challenge of accurately quantifying the healthy stage of the hippocampi. To address temporal variations and abnormalities, we averaged over a high number of time-windows (500), with each group sampled from different recording times.

In estimating velocities, we employed Euclidean distances between electrodes as a proxy for transferred distances. However, it is important to note that the actual propagation distances are likely to be longer than the Euclidean distances for two primary reasons. Firstly, neural pathways do not follow straight lines, and secondly, while pathways are conventionally

defined as unidirectional, they may only be effectively unidirectional with some uneven bidirectional flow. Consequently, the velocities we obtained may be underestimated.

## Conclusion

In summary, utilizing nonlinear coherence analysis, which inherently assumes effective unidirectional flow along its pathways, applied to data collected along the long axes of the hippocampus with its expected one-dimensional flow, we present evidence indicating that the coherence frequency of traveling waves along the hippocampus encodes the temporal aspects of communication. Specifically, velocities at low frequencies ($<\sim35$Hz) exhibit a linear dependency on frequency with a spatial frequency of approximately 50 1/m, while at higher frequencies, the velocities show stronger dependence on frequency. These sharp transitions in velocities suggest different underlying mechanisms. Although these findings were observed within the hippocampus, we propose that they are applicable to other cortical areas as well.

## Supporting Information

**S1 Fig. Merged MRI and CT Images of Subject 1, Left View.** The red dot indicates contact number 5 on the SEEG electrode inserted to the left hippocampus.
(TIF)

**S2 Fig. Merged MRI and CT Images of Subject 1, Right View.** The red dot indicates contact number 5 on the SEEG electrode inserted to the right hippocampus.
(TIF)

**S3 Fig. Merged MRI and CT Images of Subject 2, Left View.** The red dot indicates contact number 4 on the SEEG electrode inserted to the left hippocampus.
(TIF)

**S4 Fig. Merged MRI and CT Images of Subject 2, Right View.** The red dot indicates contact number 5 on the SEEG electrode inserted to the right hippocampus.
(TIF)

**S5 Fig. Merged MRI and CT Images of Subject 3, Right View.** The red dot indicates contact number 6 on the SEEG electrode inserted to the right hippocampus.
(TIF)

**S6 Fig. Normalize numbers of pathways (left) and coherence's pairs (right) for the different hippocampi as function of frequency.** A and D represent the left and right hippocampi of subject 1, B and E represent the left and right hippocampi of subject 2, and C and F represent the right hippocampus of subject 3. Normalization was to the maximum number of pathways/pairs that is the number of combinations with all contact' signals.
(TIF)

**S1 Table. A list of all four-node pathways.** E1 to E4 are the four contact signals.
(DOCX)

## Author contributions

**Conceptualization:** Gadi Goelman.

**Data curation:** Tal Benoliel, Zvi Israel, Sami Heymann, Juan Leon.

**Methodology:** Gadi Goelman.

**Software:** Gadi Goelman.

**Writing – original draft:** Gadi Goelman, Tal Benoliel.

**Writing – review & editing:** Gadi Goelman, Tal Benoliel, Dana Ekstein.

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
