## [Decision Letter · Decision Letter 0]

20 Dec 2024

PONE-D-24-44608Velocities of Hippocampal Traveling Waves Are Proportional to Their Coherence FrequencyPLOS ONE

Dear Dr. GOELMAN,

Thank you for submitting your manuscript to PLOS ONE. After careful consideration, we feel that it has merit but does not fully meet PLOS ONE’s publication criteria as it currently stands. Therefore, we invite you to submit a revised version of the manuscript that addresses the points raised by the expert referee. In addition, the authors should compare their new method with the methods used so far and specify its specific advantage.

We look forward to receiving your revised manuscript.

Kind regards,

Stéphane Charpier

Academic Editor

PLOS ONE

Journal Requirements:

Journal requirements: When submitting your revision, we need you to address these additional requirements. 1. Please ensure that your manuscript meets PLOS ONE's style requirements, including those for file naming. The PLOS ONE style templates can be found at https://journals.plos.org/plosone/s/file?id=wjVg/PLOSOne_formatting_sample_main_body.pdf and https://journals.plos.org/plosone/s/file?id=ba62/PLOSOne_formatting_sample_title_authors_affiliations.pdf. 2. Please include a caption for figure 1, 2, 3, 4, 5 and 6. 3. Please include a caption for table 1. 4. We note that you have indicated that there are restrictions to data sharing for this study. PLOS only allows data to be available upon request if there are legal or ethical restrictions on sharing data publicly. For more information on unacceptable data access restrictions, please see http://journals.plos.org/plosone/s/data-availability#loc-unacceptable-data-access-restrictions.  Before we proceed with your manuscript, please address the following prompts: a) If there are ethical or legal restrictions on sharing a de-identified data set, please explain them in detail (e.g., data contain potentially identifying or sensitive patient information, data are owned by a third-party organization, etc.) and who has imposed them (e.g., a Research Ethics Committee or Institutional Review Board, etc.). Please also provide contact information for a data access committee, ethics committee, or other institutional body to which data requests may be sent. b) If there are no restrictions, please upload the minimal anonymized data set necessary to replicate your study findings to a stable, public repository and provide us with the relevant URLs, DOIs, or accession numbers. For a list of recommended repositories, please seehttps://journals.plos.org/plosone/s/recommended-repositories. You also have the option of uploading the data as Supporting Information files, but we would recommend depositing data directly to a data repository if possible. We will update your Data Availability statement on your behalf to reflect the information you provide. 5. Please include your tables as part of your main manuscript and remove the individual files. Please note that supplementary tables (should remain/ be uploaded) as separate ""supporting information"" files"". 6. Please include captions for your Supporting Information files at the end of your manuscript, and update any in-text citations to match accordingly. Please see our Supporting Information guidelines for more information: http://journals.plos.org/plosone/s/supporting-information. 

Reviewers' comments:

Reviewer's Responses to Questions

**Comments to the Author**

1. Is the manuscript technically sound, and do the data support the conclusions?

Reviewer #1: Partly

2. Has the statistical analysis been performed appropriately and rigorously? 

Reviewer #1: Yes

3. Have the authors made all data underlying the findings in their manuscript fully available?

Reviewer #1: Yes

4. Is the manuscript presented in an intelligible fashion and written in standard English?

Reviewer #1: Yes

5. Review Comments to the Author

Reviewer #1: This is a very interesting paper proposing a new method based on 4-node recording configurations, to detect multiple traveling waves from electrophysiological recordings. The paper is certainly worth publishing but a couple of point should be clarified.

First, it is not clear how the method compares with previous methods, in particular based on Hilbert transforms. The study of Muller et al. Nature Communications 2014 was the first to reveal traveling waves in monkey (as far as I am aware), and were demonstrated there with voltage-sensitive dye (VSD) imaging. In that study they used Hilbert transforms to detect traveling waves from the data and represent the data with phase maps, where traveling waves appear more naturally. (BTW they had an instantaneous frequeny around 10Hz, and velocity about 0.5 ms, which I think matches your estimates). What do you get if you calculate the phase map with the 4 nodes? Should you not get the same information? Please clarify how your method differs from using Hilbert phase, or how they are related.

Second, regarding the physiological role of traveling waves, the same authors as above (Chemla et al. J Neurosci 2020) showed that creating "collisions" between traveling waves in monkey V1 reveals a suppressive component. This suppression was found to help discriminating ambiguous stimuli, so here the waves seems to augment visual acuity. I think this possible role for traveling waves should be mentioned at least. The question is whether you can infer such suppressive (or amplifying) interactions between multiple waves using your method, can you please clarify?

If those points are clarified I would highly recommend publication.

6. PLOS authors have the option to publish the peer review history of their article (what does this mean? ). If published, this will include your full peer review and any attached files.

**Do you want your identity to be public for this peer review?** For information about this choice, including consent withdrawal, please see our Privacy Policy .

Reviewer #1: No

---

## [Author Response · Author response to Decision Letter 1]

2 Jan 2025

We thank the editor and the reviewer for their valuable critiques and comments. Below, we provide our responses and describe the changes made in accordance with these suggestions:

Editor Comments

Please include a caption for figures 1, 2, 3, 4, 5, and 6.

Captions for these figures have been added at the end of the manuscript.

Please include a caption for table 1.

A caption for table 1 has been added in the revised version. Thank you.

Restriction on data sharing:

The local field potential (LFP) data used in this manuscript was obtained from human hippocampal recordings of epileptic patients. Due to ethical, legal, and regulatory restrictions, we have specified data-sharing availability as "Available upon reasonable request." The restrictions are as follows:

Informed Consent: Patients provided explicit informed consent for their data to be used exclusively by the Hadassah research team and not by other researchers.

Data Use Agreements (DUA): Any researcher receiving this data must sign a DUA outlining permitted uses and explicitly prohibiting unauthorized distribution or commercial use.

Research Only: The data is strictly limited to non-commercial research purposes within specific fields, such as epilepsy research.

Ethics Committee Approval: Any research involving LFP data sharing or secondary use requires prior approval from our ethics committee.

Sensitive Information: The LFP data contains details about seizure events, which makes it particularly sensitive. Consequently, restrictions limit access to seizure-related data to authorized personnel.

Potential for Re-identification: Epileptic patterns and associated clinical metadata may increase the risk of patient re-identification, necessitating additional safeguards.

Please include your tables as part of your main manuscript and remove the individual files.

This has been completed. Thank you.

Please include captions for your Supporting Information files at the end of your manuscript.

Captions for the Supporting Information files have been added at the end of the manuscript. Thank you.

Reviewer Comments

This is a very interesting paper proposing a new method based on 4-node recording configurations, to detect multiple traveling waves from electrophysiological recordings. The paper is certainly worth publishing but a couple of point should be clarified.

Thank you.

First, it is not clear how the method compares with previous methods, in particular based on Hilbert transforms. The study of Muller et al. Nature Communications 2014 was the first to reveal traveling waves in monkey (as far as I am aware), and were demonstrated there with voltage-sensitive dye (VSD) imaging. In that study they used Hilbert transforms to detect traveling waves from the data and represent the data with phase maps, where traveling waves appear more naturally. (BTW they had an instantaneous frequeny around 10Hz, and velocity about 0.5 ms, which I think matches your estimates). What do you get if you calculate the phase map with the 4 nodes? Should you not get the same information? Please clarify how your method differs from using Hilbert phase, or how they are related.

We thank the reviewer for referring us to the Muller (2014) paper. We will include this study in the revised manuscript. Notably, as the reviewer correctly mentioned, the traveling wave velocity reported in the monkey visual system in their study is close to our findings in the human hippocampus. Specifically, using our approximate expression for velocity in the frequency range examined by Muller et al. (5–20 Hz): velocity~0.02*freq, we estimate a range of 0.1–0.4 m/s. In comparison, their reported velocity range was 0.25–1.35 m/s, indicating faster flow in the visual cortex of the monkey.

In Muller et al., traveling waves were measured using a phase-based method that, in many respects, is similar to our approach. They utilized the Hilbert transform to convert the band-pass filtered time series into amplitude and phase, while we used the Wavelet transform. The Hilbert transform is commonly used for extracting instantaneous amplitude, phase, and frequency and is well-suited for narrowband signals. In contrast, Wavelet transformation decomposes a signal into time-frequency components, making it suitable for analyzing both time-localized and multi-frequency signals.

In their study, the signals were band-pass filtered in the range of 5–20 Hz, which made the Hilbert transform an appropriate and straightforward choice. In our case, however, we specifically investigated differences across multiple frequencies, for which the Wavelet transform was more appropriate due to its ability to analyze time-frequency variations.

Another significant difference between Muller’s approach and ours lies in the scope of the analysis. Muller et al. employed a bivariate (pairwise) method applied to two-dimensional data, while our method is multivariate, and was applied to one-dimensional data. We argue that to obtain accurate results, a multivariate analysis is essential, especially when analyzing pathways consisting of several nodes. Using pathways ensures that phase differences and distances along effectively unidirectional pathways are correctly calculated. In contrast, bivariate analysis could result in a mixture of coherences that biases the phase and distance estimations, leading to incorrect velocity calculations. This point is extensively discussing in the paper to explain the difference in estimated velocities between bivariate and multivariate analyses.

However, when using two- (or three-) dimensional data, it is possible to correctly identify traveling waves, and thus to calculate the phase and distances correctly. This is since the observation of 2D traveling waves inherently accounts for the coherence mixture. Thus, the analysis in Muller et al.'s paper is valid and its estimated velocity is similar to ours.

Furthermore, a more recent study by Mohan et al. (Nat Hum Behav, 2024), cited in our manuscript, used a similar 2D-like approach with EEG data from the human cortex. Their velocity estimates align closely with ours, reinforcing the reliability of our methodology.

Second, regarding the physiological role of traveling waves, the same authors as above (Chemla et al. J Neurosci 2020) showed that creating "collisions" between traveling waves in monkey V1 reveals a suppressive component. This suppression was found to help discriminating ambiguous stimuli, so here the waves seems to augment visual acuity. I think this possible role for traveling waves should be mentioned at least. The question is whether you can infer such suppressive (or amplifying) interactions between multiple waves using your method, can you please clarify?

We agree that it is appropriate to include in the introduction the role of traveling waves in addressing the "correspondence problem." This addition will be made—thank you for the suggestion.

Regarding the use of multivariate pathway analysis to test the “correspondence problem”, this requires further consideration and experimental validation. As noted earlier, the strengths and limitations of our approach lie in its ability to characterize nonlinear, continuous interactions that define pathways of information flow, while acknowledging that these pathways are inherently one-dimensional.

We will seek data where two stimuli (either simultaneously or sequentially) are expected to initiate synchronous activity at distinct locations, propagate along pathways, and converge at certain points. These meeting points can be conceptualized as hubs that initiate new pathways or as terminus points for the pathways.

Our initial approach to investigate this will involve a straightforward occurrence strategy: calculating the probability of different locations (ROIs) based on their positions within the pathways (e.g., first, second, third, or fourth) and calculating their coherence strength.

---

## [Decision Letter · Decision Letter 1]

16 Jan 2025

Velocities of Hippocampal Traveling Waves Are Proportional to Their Coherence Frequency

PONE-D-24-44608R1

Dear Dr. GOELMAN,

We’re pleased to inform you that your manuscript has been judged scientifically suitable for publication and will be formally accepted for publication once it meets all outstanding technical requirements.

Kind regards,

Stéphane Charpier

Academic Editor

PLOS ONE

Additional Editor Comments (optional):

Reviewers' comments:

Reviewer's Responses to Questions

**Comments to the Author**

1. If the authors have adequately addressed your comments raised in a previous round of review and you feel that this manuscript is now acceptable for publication, you may indicate that here to bypass the “Comments to the Author” section, enter your conflict of interest statement in the “Confidential to Editor” section, and submit your "Accept" recommendation.

Reviewer #1: All comments have been addressed

2. Is the manuscript technically sound, and do the data support the conclusions?

Reviewer #1: Yes

3. Has the statistical analysis been performed appropriately and rigorously? 

Reviewer #1: Yes

4. Have the authors made all data underlying the findings in their manuscript fully available?

Reviewer #1: Yes

5. Is the manuscript presented in an intelligible fashion and written in standard English?

Reviewer #1: Yes

6. Review Comments to the Author

Reviewer #1: Thank you for improving the paper, all my remarks have been addressed appropriately and I have no further remarks.

7. PLOS authors have the option to publish the peer review history of their article (what does this mean? ). If published, this will include your full peer review and any attached files.

**Do you want your identity to be public for this peer review?** For information about this choice, including consent withdrawal, please see our Privacy Policy .

Reviewer #1: No

---

## [Editor Report · Acceptance letter]

PONE-D-24-44608R1

PLOS ONE

Dear Dr. GOELMAN,

I'm pleased to inform you that your manuscript has been deemed suitable for publication in PLOS ONE. Congratulations! Your manuscript is now being handed over to our production team.

Kind regards,

on behalf of

Pr. Stéphane Charpier

Academic Editor

PLOS ONE
